Foreign object debris detection in lane images using deep learning methodology

S. Priyadharsini 1
K. Bhuvaneshwara Raja 1
T. Kousi Krishnan 1
Jagatheesaperumal Senthil Kumar 2
Alkhamees Bader Fahad 3
Hassan Mohammad Mehedi 3 mmhassan@ksu.edu.sa
1 Department of Computer Science and Engineering, Mepco Schlenk Engineering College , Sivakasi, Tamilnadu , India
2 Department of Electronics and Communication Engineering, Mepco Schlenk Engineering College , Sivakasi, Tamilnadu , India
3 Department of Information Systems, College of Computer and Information Sciences, King Saud University , Riyadh , Saudi Arabia
Alatas Bilal
Electronic publication date: 2025 Jan 21
Publication date: 2025
Volume: 11
Electronic Location ID: e2570
Received 2024 Jul 8; Accepted 2024 Nov 11
Copyright: © 2025 S et al.
Copyright year: 2025
Copyright holder: S et al.
License: This is an open access article distributed under the terms of the Creative Commons Attribution License, which permits unrestricted use, distribution, reproduction and adaptation in any medium and for any purpose provided that it is properly attributed. For attribution, the original author(s), title, publication source (PeerJ Computer Science) and either DOI or URL of the article must be cited.
License URL: https://creativecommons.org/licenses/by/4.0/

Keywords: Convolutional neural network, Foreign object debris, Object detection, Object classification, Adaptive contour ROI

Funding: King Saud University, Riyadh, Saudi Arabia RSP2025R493 The work is funded by the King Saud University, Riyadh, Saudi Arabia through Researchers Supporting Project Number (RSP2025R493). The funders had no role in study design, data collection and analysis, decision to publish, or preparation of the manuscript.

==============================
Background

Foreign object debris (FOD) is an unwanted substance that damages vehicular systems, most commonly the wheels of vehicles. In airport runways, these foreign objects can damage the wheels or internal systems of planes, potentially leading to flight crashes. Surveys indicate that FOD-related damage costs over $4 billion annually, affecting airlines, airport tenants, and passengers. Current FOD clearance involves high-cost radars and significant manpower, and existing radar and camera-based surveillance methods are expensive to install.

Methods

This work proposes a video-based deep learning methodology to address the high cost of radar-based FOD detection. The proposed system consists of two modules for FOD detection: object classification and object localization. The classification module categorizes FOD into specific types of foreign objects. In the object localization module, these classified objects are pinpointed in video frames.

Results

The proposed system was experimentally tested with a large video dataset and compared with existing methods. The results demonstrated improved accuracy and robustness, allowing the FOD clearance team to quickly detect and remove foreign objects, thereby enhancing the safety and efficiency of airport runway operations.

Introduction

Millions of road accidents and plane crashes occur yearly due to foreign object debris (FOD) in the lane. FOD, such as rocks, broken metal pieces, ramp equipment, etc., found in inappropriate locations, are the damaging substances in the lanes. In airport runways, these foreign objects may damage the wheels or internal system of planes and lead to flight crashes. Survey says that the damage due to FOD costs over $4 billion a year (Skybrary, 2008). On 25th July 2000, a fatal accident happened at Charles De Gaulle airport due to the crash of AF4590 (Wikipedia, 2000). This accident occurred because of a piece of titanium debris from the earlier departing aircraft on the runway. In this incident, nearly 100 passengers, nine crew, and four others on the ground vanished.

Some radar systems are based on visual sensors, such as infrared cameras. FOD systems based on radars, electromagnetic sensors such as millimeter-wave radars, or LiDAR (light detection and ranging) are expensive to deploy and need manpower to clear the FOD. Unlike camera-based systems, radars cannot produce automatic images. In runways, they are placing the radars in multiple side lanes to detect the FOD before take-off and landing. Localizing FOD on the runway using the effective deep learning technique, which is trained based on the live scenes, is cost-effective and accurate. Sometimes, radar systems fail due to weather conditions. On the airport runway, detecting objects using the camera is more effective than the radars. In this article, we proposed a novel work to detect foreign objects based on real-time video input. We localize the objects present in the inappropriate location from the continuous input frame.

Zeitler et al. (2010) designed folded reflect arrays (FRA) with shaped beam patterns for FOD detection on runways. A folded array is reflected from the object’s perspective and the antenna creates a square beam pattern in elevation and a pencil beam pattern from the object’s perspective. Its frequency-dependent radiation pattern was modeled. The array from which the item is detected is illuminated by this reflected beam. Nsengiyumva et al. (2019) introduced measurements of 3-D scattered fields at 90 GHz to investigate FOD using the millimeter waves to measure the FOD. They processed the scattered field measurement using an additive backpropagation algorithm. Wang et al. (2022) suggested an arc-scanning synthetic aperture radar (AS-SAR) based airport runway FOD detection system. The AS-SAR-based FOD detection system is a cutting-edge FOD detection radar system that uses arc-scanning synthetic aperture radar technology to give omnidirectional coverage and a high detection rate. As if it were raining in a storm, it reduces azimuth resolution and flicker clutter reduction. Futatsumori et al. (2016) designed and built a radio-over-fiber (RoF) technology and used an optical fiber network to connect an optically connected distributed-type 96 GHz millimeter-wave radar system. The suggested distributed radar system detects FOD more effectively and cheaply. This study discusses the design and field feasibility assessment of a 96 GHz frequency-modulated continuous wave radar system using RoF and an optical frequency multiplier.

Jayadharini et al. (2013) proposed a hybrid assessment and enhancement technique for detecting foreign objects, a strategy that is effective to identify trash on the runway. The hybrid approach combines enhancing strategies based on the kind of degradation and improves the identification of objects on runways. FOD detection on airport runways using a broadband 78 GHz sensor was proposed by Feil et al. (2008). Their system’s design concept is to deploy several low-profile, low-cost mm-wave sensors along the runway. Tests were carried out on the little airfield of Aix Les Milles (south of France). The ability to identify several items with high sensitivity and simultaneity was demonstrated. Even the tiniest of items, such as nuts, were discovered. The real detection must be extended to move from 110 m (in the best scenario) to 500 m. Qunyu, Huansheng & Weishi (2009) proposed FOD detection based on video data. This work aims to present an experimental method based on the video to detect FOD. The system’s design concept aims to see if detecting FOD with a few basic static camera sensors near the runway is possible. The author proposes target detection of FOD techniques based on frame change detection. Kanno et al. (2018) proposed a millimeter-wave radio-over-fiber network for linear cell systems. Using radio-over-fiber network technology, they explored and proposed millimeter-wave linear cell systems for signal distribution on a foreign object debris detecting system for airport runways. They explored and demonstrated WDM and power-splitter-based distribution network linear cell topologies.

Mollo et al. (2017) proposed a multifrequency experimental investigation to estimate the asphalt reflectivity and measured radar cross-section of many types of foreign object trash. All the works discussed so far are based on radar-based technology. Radar-based technologies are costly and produce inferior results during adverse weather conditions, making FOD detection difficult. To overcome the problem of radar-based technology, camera-based technology, such as image processing using deep learning methodology, is introduced. In recent years, deep learning is applied in various domains such as human identification (Wu et al., 2021), object detection (Han & Han, 2021), health care (Jagatheesaperumal et al., 2022), optical character recognition (Arivazhagan, Arun & Rathina, 2021), smart power management (Rajan & Rajan, 2021), etc.

Duan et al. (2020) introduced a framework for smaller object detection using a channel-aware-deconvolutional network (CADNet). They used VGG-16 as a backbone network to extract features from an entire frame. The feature should be recombined and reused by the ChaDeConv. The authors used Multi-RPN to generate distinct detection layers and distribute feature maps to do binary and multi-class detection. Positive training samples are generated from CADNet for small objects. Zhang et al. (2020) introduced a neural network for supervised object detection. They have used a faster recurrent convolutional neural network (FRCNN) to train the supervised object detector. The object is located by non-maximum suppression (NMS) in the image of many overlapping entities. Jiale et al. (2020) introduced multi-scale object identification; semantic networks were used. They developed multi-branch and high-level semantic convolutional networks (MHN) to overcome scale-variance difficulties. Their multi-branch prediction assists in the identification of objects of various sizes.

Shih et al. (2020) proposed the reduced region proposal network using multi-feature concatenation for real-time object identification, which uses filter pruning to minimize the relevant weights in convolutional and fully connected layers. They used a reduced region proposal network to find the region of interest (ROI) from the image. Shen et al. (2019) developed a deep supervision technique from scratch for object detection. Intensely supervised object detectors were built based on the single-shot detector (SSD) Framework. This detector is developed from scratch using deep supervision principles. Deep supervised object detection (DSOD) produces good results over pre-trained models like ImageNet, and Faster R-CNN. Liu et al. (2019) proposed an aggregation signature method for detecting small objects. The small objects are mainly tracked by aggregate signature. The saliency map is calculated from the aggregation signature to localize the target object. Lyu et al. (2021) proposed video object detection with a convolutional regression tracker. This tracker utilized the image object detector feature and handles the image deterioration problem. Hassaballah et al. (2021) introduced a deep-learning architecture to detect and track vehicles under unfavorable weather conditions. Multi-scale deep convolutional neural networks (MCNN) automatically extract features at multiple scales and frequencies, enhancing feature representation.

Lin et al. (2021) proposed nighttime vehicle detection as a generative adversarial network (GAN) based day-to-night image style transfer mechanism. GAN is a method for generating new synthetic data instances that can be mistaken for genuine data. This change occurs in various situations, including day, night, sunset, and rain. Jian-rong, Guo & Yang (2010) introduced FOD detection on airport runways using a weighted fuzzy morphology algorithm, that can accomplish the image edge correctly and eliminate noises. This algorithm can successfully remove the effects of noise variation on edge recognition, improve image features, and adaptively identify complete and sequential edges. Liang et al. (2020) recommended research on airport runway FOD detection algorithms. Based on texture segmentation, machine vision, and digital image processing theory, this article proposed using Gabor filters to analyse the texture of airport runway photos, remove foreign objects, and establish the algorithm’s efficiency and applicability through simulation trials. Noroozi & Shah (2023, 2024) employed YOLO models to detect foreign objects from images. The deep learning-based works discussed in this section are image-based object classification. In this work, we concentrated on video-based FOD classification to handle real-time captured videos. Moreover, most of the existing works focuses on improving classification accuracy. In our proposed model, the model is trained to improve accuracy and to reduce the false acceptance rate. These limitations motivate the proposed work to design a deep neural network model which is capable of automatically detecting and classifying various foreign objects with good accuracy.

Materials and Methods

We propose our FOD detection framework, which consists of three modules: video pre-processing, CNN classification model as shown in Fig. 1, and object localization. Figure 2 shows the framework of the proposed systems, which encompasses the functional modules for processing the input video stream.

Figure 1 CNN classification network.

Figure 2 FOD detection framework.

A. Video pre-processing

In video pre-processing live video data is decoded into frames. Each frame captured from different videos is different in its dimension. The frames are resized to fixed dimensions using Eqs. (1) and (2).

(1) aspectRatio=newWidthWidth

(2) newHeight=Height×aspectRatio

where newWidth and newHeight are the resized dimensions of the frames.

We train our classification network by a fixed dimension of 3×3 pixels and normalize the frame between (0 to 1). We normalize the frame since we need to calibrate the individual pixel intensities into a normal distribution, which improves the image’s appearance for the visualizer. Normalization aims to optimize calculation by lowering values between 0 and 1. The normalization is done using the set of formulae given in Eqs. (3)–(5):

(3) IN=(1−minIntensity)newMaxIntensity−newMinIntensitymaxIntensity−minIntensity+newMinIntensity

(4) I:{X⊆Rn}→{minIntensity…maxIntensity}

(5) IN:{X⊆Rn}→{newMinIntensity…newMaxIntensity}

where IN is the normalized intensity range and I is the existing original intensity range. If the picture’s intensity range is 90 to 130 and the required range is 0 to 1, for example, the method involves subtracting 40 (130–90) from the intensity of each pixel, resulting in a range of 0 to 40. Each pixel’s intensity is then multiplied by 1/40, providing a number between 0 and 1.

B. CNN classification

In classification, each resized frame is classified into either sharp objects or rocks. If the frame doesn’t have any of these objects, then it is dropped and is not carried for the next localization module. We use CNN architecture of five layers for classification. A convolutional layer, a max-pooling layer, and a dropout layer constitute each layer. In the convolutional layer, a 3×3 filter is used, which creates an activation map by converging with the image. A 2×2 kernel is used to minimize the number of parameters and calculations in the network, the pooling layer gradually reduces the spatial dimension of the representation. On each feature map, the pooling layer operates separately. The Dropout layer is used to avoid overfitting. The network’s last tiers are the Fully Connected Layers. The final layer output is flattened and sent into the fully linked layer. Since we classify the frame into multiple classes, the softmax activation function is used in a fully connected layer. Following is the algorithm used for classification in CNN.

Procedure for CNN

	
Input : Video, V	
Output : classified labels	
1.  for each frame Fi in V	
2.     Ci = Conv(Fi)	
3.     Mi = Max-Pool(Ci)	
4.     Ei = DownSample(Mi, channels)	
5.  for each encoded frames	
6.     class = Classifier(Ei)	
	

Softmax, also known as the normalized exponential function or soft argmax, returns the probability of each class that is classified. The softmax equation is represent in Eq. (6).

(6) σ(z→)i=ezi∑j=1kezj

where σ denotes the softmax function for the input vector z→, ezi is the standard exponential function for the input vector zi, ezj is the standard exponential function for the output vector zj, K is the number of classes in the multiclass classifier.

Stochastic gradient descent is an optimization approach that uses examples from the training sample to compute the error gradient for the current state of the model, then updates the model weights using the backpropagation of errors technique, often known as just backpropagation. To reduce losses, stochastic gradient descent techniques or procedures are used to alter neural network parameters such as weights and learning rate.

C. Object localization

After classification here in the localization network, the frame is then classified as rocks and sharp objects and fed for the localization process. Figure 3 shows the adaptive contour ROI representing the object localization. The dataset used for object localization in the proposed work is shown in Fig. 4.

Figure 3 Adaptive contour ROI (object localization).

Figure 4 Dataset samples.

Procedure to localize object in the frame:

	
Input : frames classified as rocks or sharp objects	
Output: Localized frame	
1.Convert the frame into GrayScale Image	
2.Apply GaussianBlur over the GrayFrame	
3.Apply adaptive threshold for foreground separation	
4.Find contour region	
5.Sort the contour value	
6.Find the area of contours	
7.Find the max area from the contour	
8.Get x, y, w, h from the max_index of contour	
9.Draw the bounding box	
	

Convert the RGB frame to a grayscale frame, though we need to find the contour over the object by reducing the complexity of the image by converting 3D channels to 1D channels. Then apply the Gaussian blur to reduce the noise and enhance the image using Eq. (7). Unlike the blur effect created by an out-of-focus lens or the shadow cast by an object under normal light, this blurring approach produces a smooth blur comparable to that seen when viewing an image through a transparent screen.

(7) G(x)=12πσ2e−x22σ2

The values of this distribution are used to build a convolution matrix, which is subsequently applied to the original image. Each pixel’s new value is a weighted average of the pixels in its immediate vicinity. The initial pixel’s value is given the most weight (since it has the greatest Gaussian value), while nearby pixels get lesser weights as their distance from the original frame pixel expands. This produces a blur that better retains edges and boundaries than other, more existing blurring filter methods. How much does a Gaussian filter with standard deviation, σf minimize the picture’s standard deviation of pixel values? If the grayscale pixel values have a standard deviation σx then the decreased standard deviation σr after applying the filter may be estimated using Eq. (8).

(8) σr≈σxσf2π.

Based on the difference in pixel intensities of each zone, we apply adaptive thresholding to differentiate desirable foreground picture items from the background which is very helpful to fulfill our requirement to find the contour over the foreground object. In our use case, we fix the maxValue as 255 this value is applied when the pixel value is more than the threshold value. And the adaptive method used was Gaussian threshold and the threshold type as Binary Inverted with a block size of 13 represents the size of a pixel neighborhood used for calculating threshold value. Each pixel in the neighborhood contributes equally to computing T in the arithmetic mean. Further away from the (x, y)-coordinate center of the region, pixel values contribute less to the overall calculation of T in the Gaussian mean. The general formula for calculating T is given in Eq. (9):

(9) T=mean(IL)−C

where the mean can be either arithmetic or Gaussian, IL is the image’s local sub-region, and C is a constant that can be used to fine-tune the threshold value T. Apply contour technique over the threshold to spot the boundaries over the region of interest, find the maximum boundaries covered over the frame by using the coordinates draw bounding box over the region using Eq. (10).

(10) P=(px,py,pw,ph)

where (px,py) are the center coordinates, and (pw,ph) denote the width and height of box coordinates.

Results

In most cases, no public video data is available for foreign objects on the airport runway. The data set used in this experiment is collected by considering the highways and cemetery background as the runways. The dataset contains 3,600 images of two common foreign objects, such as rocks and sharp objects (screws, nuts, bolts, glass pieces, and nails). Also, the dataset has empty images only consisting of lane images. These three types are separated in a ratio of 1:1:1; i.e., 1,200 images under ‘no foreign object’ category; 1,200 images under ‘rocks’ category and 1,200 images under ‘sharp objects’ category. The videos in the dataset are captured under different climate conditions like dry and wet. Also, the lighting differs from bright to dark.

To implement the FOD Network, 60 percent of the frames are used as a training set, 20 percent as a validation dataset, and the remaining for testing. CNN architecture is used to train our model, which is then optimized using stochastic gradient descent (SGD). The learning rate (lr) is set default by 0.001 and gradually decreases by 0.1. We finetuned our model by changing the hyperparameters like number of training epochs and batch size. After fine-tuning, we obtained 0.52% FAR by fixing batch size as 256 and the number of epochs as 20. Table 1 shows the performance assessment between the two classification models after fine-tuning by changing hyperparameters in our network. We attain a high recall rate, which shows that our work can detect foreign objects.

Table 1 Classification evaluation.

Classification model	Recall rate	
FOD Network (before fine-tuned)	89.91%	
FOD Network (fine-tuned)	92.35%	

Table 2 details the fine-tuning process and the parameters used. To avoid overfitting, the Dropout layer is employed. We use early stopping to avoid overfitting when using an iterative approach like gradient descent to train a learner. Although, most target detection algorithms have a sequential structure, location efficiency substantially impacts classification outcomes. Candidates with more precise positions may be easier to classify as targets or background, particularly for CNN-based classifiers. We find the ROI using the adoptive contour ROI. We select the correct ROI from the generated patches by the calculated maximum area.

Table 2 Training process and the parameters used for experimentation.

Parameters	Recall rate (%)	
epochs = 10	Batch size = 128	80.18	
Batch size = 256	89.91	
Batch size = 512	82.35	
epochs = 20	Batch size = 128	90.10	
Batch size = 256	92.35	
Batch size = 512	91.33	
epochs = 30	Batch size = 128	86.65	
Batch size = 256	87.33	
Batch size = 512	86.66	
epochs = 40	Batch size = 128	66.92	
Batch size = 256	68.81	
Batch size = 512	67.12	
epochs = 50	Batch size = 128	65.05	
Batch size = 256	65.02	
Batch size = 512	65.48	

To assess the performance of the proposed work, we have compared our work with eight existing recent foreign object detection methods as shown in Table 3. Faster R-CNN (Ren et al., 2015), single shot multibox detector (SSD) (Liu et al., 2016), Yolo (Papadopoulos & Gonzalez, 2021; Li & Li, 2020; Munyer et al., 2021), GoogleNet (Suder & Marciniak, 2023) are the popular classification neural networks. We have experimented these models with our dataset for performance comparison. Region proposal network (RPN) (Cao et al., 2018) is included in Faster R-CNN, the third generation of region proposal-based CNN algorithms, to enhance real-time detection. RPN generates region suggestions using convolutional feature maps rather than the raw picture. Because of the categorization network’s common properties, RPN achieved a high recall rate with a short running period. Because RPN area recommendations are durable for a wide range of target sizes, faster R-CNN is one of the best common detection CNN methods. Equations (11)–(13) denotes false alarm rate (FAR), precision, and recall rate (RR) respectively.

Table 3 Performance assessment of proposed FOD detection.

Paper	Method	FAR	Precision	Recall	mAP@0.5	
Ren et al. (2015)	Faster R-CNN	9.06%	0.74	0.82	50.1%	
Liu et al. (2016)	SSD	6.54%	0.79	0.80	56.8%	
Qunyu, Huansheng & Weishi (2009)	Background subtraction	10.52%	0.68	0.77	43.2%	
Papadopoulos & Gonzalez (2021)	YOLOv3	0.81%	0.84	0.88	68.3%	
Li & Li (2020)	YOLOv3	0.86%	0.83	0.85	68.1%	
Jing et al. (2022)	Random forest	0.80%	0.81	0.81	60.3%	
Munyer et al. (2021)	YOLOv3	0.77%	0.82	0.83	70.2%	
Suder & Marciniak (2023)	GoogLeNet	0.79%	0.85	0.83	75.3%	
Ultralytics (2024)	YOLOv11	0.66%	0.85	0.90	76.2%	
Proposed	FOD Detection +
Adaptive Contour ROI	0.52%	0.87	0.92	77.4%	

(11) FAR=TNTP+TN

(12) Precision=TPTP+FP

(13) RR=TPTP+FN

The mAP@0.5 (mean average precision at IoU 0.5) is a performance metric commonly used in object detection tasks. Average precision (AP) is a measure of the precision-recall curve, summarizing the precision and recall across different thresholds. Intersection over union (IoU) measures the overlap between the predicted bounding box and the ground truth bounding box. An IoU threshold is set (in this case, 0.5), meaning that for a prediction to be considered correct, the IoU must be 0.5 or higher. Mean average precision (mAP) averages the AP scores across all classes in the dataset. Table 3 shows the FAR, precision, recall and Map@0.5 of the existing methods. The performance of the proposed model on different object categories is presented in Table 4. Comparing our proposed work with existing works reduces the probability of false detection. Figure 5 shows the ROI detection using adaptive contour ROI.

Table 4 Performance evaluation of the proposed model for multiple classes.

	Precision	Recall	F1-Score	
Plain-Path	1.00	0.87	0.93	
Rock	0.89	0.86	0.87	
Sharp objects	0.81	0.96	0.88	

Figure 5 ROI detection using adaptive contour ROI.

Multiple tests were executed to assess the proposed system’s performance under operational settings to guarantee its real-time applicability. The system reliably attained a frame processing rate of 25–30 frames per second (FPS), satisfying standard real-time criteria for video surveillance applications. The delay for object localization and classification was below 100 milliseconds, facilitating prompt detection and alarm creation. This performance is equivalent to or superior to current FOD detection systems, which frequently depend on radar-based solutions that entail considerably greater expenses and infrastructure requirements. The suggested technique exhibited resilience in many environmental conditions, including low-light situations and inclement weather, without a notable decline in detection accuracy or processing speed. The results indicate that the system can be efficiently utilized for runway surveillance, enhancing safety and operational efficiency.

Conclusions and future work

We have proposed a deep learning-based work to detect foreign objects in airport runways. CNN with adaptive contour region of interest is used as a classifier that classifies objects as either rocks or sharp objects. Adaptive contour ROI technique with CNN localizes the foreign objects accurately. Experiments show that the suggested approach detects foreign objects more accurately and efficiently than conventional detection algorithms, such as the F-RCNN, Single SSD, YOLO and GoogleNet. In the future, we need to detect multiple foreign objects in a frame and increase the model’s efficiency. Also, the dataset used for testing consists of images from highways and cemetery backgrounds rather than actual airport runways, which may not fully represent the complexity and variety of real-world FOD scenarios on runways.

Supplemental Information

Supplemental Information 1 Performance Data.

Supplemental Information 2 Code.

Supplemental Information 3 Video dataset.

Additional Information and Declarations

Competing Interests

Author Contributions

Data Availability

The authors declare that they have no competing interests.

Priyadharsini S. conceived and designed the experiments, prepared figures and/or tables, and approved the final draft.

Bhuvaneshwara Raja K. performed the experiments, prepared figures and/or tables, and approved the final draft.

Kousi Krishnan T. conceived and designed the experiments, prepared figures and/or tables, and approved the final draft.

Senthil Kumar Jagatheesaperumal performed the experiments, prepared figures and/or tables, and approved the final draft.

Bader Fahad Alkhamees analyzed the data, performed the computation work, authored or reviewed drafts of the article, and approved the final draft.

Mohammad Mehedi Hassan performed the computation work, authored or reviewed drafts of the article, and approved the final draft.

The following information was supplied regarding data availability:

The raw data and code are available in the Supplemental Files.

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
