# Peer review of "Foreign object debris detection in lane images using deep learning methodology"

_PeerJ Computer Science, doi:10.7717/peerj-cs.2570_

## Round 0.1 · original submission · Major Revisions

Dear authors,

Thank you for submitting your article. Based on reviews' comments, your article has not yet been recommended for publication in its current form. However, we encourage you to address the concerns and criticisms of the reviewers and to resubmit your article once you have updated it accordingly. Before submitting the paper, following should also be addressed:

1. Equations should be used with correct equation number. Please do not use “as follows”, “given as”, etc. Explanation of the equations should also be checked. All variables should be written in italic as in the equations. Their definitions and boundaries should be defined. Necessary references should also be provided.
2. Many of the equations are part of the related sentences. Attention is needed for correct sentence formation.
3. English grammar and writing style errors should be corrected.
4. Some paragraphs are too long to read. They should be divided into two or more for readability and comprehensibility.
5. Extensive experiments are required to demonstrate the efficiency and applicability of the proposed method.
6. The values for the parameters of the used method should be provided.
7. Pay special attention to the usage of abbreviations. Spell out the full term at its first mention, indicate its abbreviation in parenthesis and use the abbreviation from then on.
8. Some more recommendations and conclusions should be discussed about the paper considering the experimental results. The conclusion section is weak. There is also no discussion section about the results. It should briefly describe the results of the study and some more directions for further research. You should describe the academic implications, practical advantages, main findings, research limitations and directions for future research in the conclusion section. The conclusion in its current form is generally confused. What will be happen next? What we supposed to expect from the future papers? So rewrite it and consider the following comments:
- Highlight your analysis and reflect only the important points for the whole paper.
- Mention the benefits
- Mention the implication in the last of this section.
9. Algorithms should be provided in a same single format.
10. References should be written according to single PeerJ referencing style. They are untidily written in the list.
11. Figures should be polished.
12. Reviewers have advised you to provide specific reference/data. You are welcome to add them if you think they are useful and relevant. However, you are under no obligation to include them, and if you do not, it will not affect my decision.

Best wishes,

Reviewer 1 ·

Basic reporting

1) Language Improvement:
Avoid using contractions like "don’t" (line 147). It should be replaced with "do not". Improving the language of the manuscript is highly recommended for clarity and professionalism.

2) Citation Requirement:
Proper citation is required for all statements. For example, the statement “Surveys indicate that FOD-related damage costs over $4 billion annually, affecting airlines, airport tenants, and passengers” needs a proper citation.
Identify all similar statements in the manuscript and ensure they are properly cited.

Experimental design

1) Result Section - Dataset Availability:
The authors mentioned that there is no public FOD dataset. Please review the available dataset and use it if it is applicable and useful for the study.
2) Comprehensive Comparison:
There is a need for a comprehensive comparison of the proposed method with existing methods in the literature.
3) Dataset Information:
Provide more detailed information regarding the dataset used (e.g., quantity per category, number of backgrounds used, etc.).
4) Fine-Tuning Process:
Create a table detailing the fine-tuning process and the parameters used.

Validity of the findings

1) Literature Inclusion and Comparison:
Please review the materials in the following two papers and include them in the literature if they are relevant:
a. "Towards Optimal Foreign Object Debris Detection in an Airport Environment"
b. "Open-World Foreign Object Debris Detection Framework Empowered by Generative Adversarial Network and Computer Vision Models"

If these papers are highly relevant, a comparative analysis is necessary.
2) Limitations and Shortcomings:
The authors have mentioned: “The deep learning-based works discussed in this section lack precision/recall rate for object classification. Moreover, the existing works do not classify foreign objects into multiple categories like rock, sharp objects, etc.”
However, the literature contains numerous works with the metrics discussed by the authors. Refer to the comparison section of “Open-World Foreign Object Debris Detection Framework Empowered by Generative Adversarial Network and Computer Vision Models” for more information.
The shortcomings of the literature should be elaborated comprehensively, considering the existing works with relevant metrics.

Additional comments

Please talk about future research.

Cite this review as

Reviewer 2 ·

Basic reporting

/

Experimental design

/

Validity of the findings

/

Additional comments

The manuscript proposes a deep learning-based method for detecting and classifying FOD in lane images, specifically on airport runways, to improve safety and efficiency in FOD clearance operations. The proposed method consists of three modules: video pre-processing to resize and normalize frames, CNN classification to categorize frames into specific types of foreign objects, and object localization using an adaptive contour ROI technique to pinpoint objects in the frames.

1. The dataset used for testing consists of images from highways and cemetery backgrounds rather than actual airport runways, which may not fully represent the complexity and variety of real-world FOD scenarios on runways.

2. The classification is limited to only two types of foreign objects—rocks and sharp objects. This narrow focus might reduce the method’s applicability to other types of FOD that could also be dangerous: https://github.com/FOD-UNOmaha/FOD-data

3. The manuscript compares its method primarily with two existing approaches (Faster R-CNN and SSD), but it does not include other relevant FOD-focused methods that can provide a more comprehensive evaluation of its performance.

4. Although the paper mentions efficiency, it would benefit from a more detailed discussion on the method’s real-time applicability, especially considering the high demands of runway safety operations.

5. The paper lacks a detailed discussion on how well the proposed method generalizes across different environments and lighting conditions, which are crucial for practical deployment.

6. The paper could be strengthened by including ablation studies that assess the contribution of each module or component of the proposed method to the overall performance.

Cite this review as

---

## Round 0.2 · Minor Revisions

Dear authors,

Thank you for the revision. One of the previous reviewers did not respond to the invitation for the revised paper. Although your paper seems improved, please make necessary additions and modifications advised by Reviewer1. By the way, many of the equations are part of the related sentences and attention is needed for correct sentence formation. Furthermore, figures should be polished. Some of them have low resolution.

Best wishes,

Reviewer 1 ·

Basic reporting

no comment

Experimental design

1) In Table 3, the authors compare their proposed model with several existing object detection models based on False Alarm Rate (FAR). To provide a more comprehensive evaluation, the authors are encouraged to include additional metrics for all models, specifically [email protected], precision, and recall.
2) The authors are requested to evaluate their dataset using the latest version of the YOLO model (likely version 11, though the authors should verify this) and report the results in Table 3.
3) It would be valuable for the authors to present the performance of the proposed model on different object categories separately, with a dedicated table showing results for each class.

Validity of the findings

no comment

Additional comments

no comment

Cite this review as

---

## Round 0.3 · Minor Revisions

Dear authors,

Thank you for the revision. Although you write "The performance of the proposed model on different object categories is presented in Table 4" according to respected Reviewer1's concerns and criticisms, there is not Table 4 in the manuscript. It is requested that the performance of the proposed model on different object categories be presented separately, with a dedicated table showing results for each class, as previously requested.

Best wishes,

Reviewer 1 ·

Basic reporting

no comment

Experimental design

no comment

Validity of the findings

no comment

Additional comments

no comment

Cite this review as

---

## Round 0.4 · accepted · Accept

Dear Authors,

I am grateful to you for addressing the concerns and criticisms of the reviewer in a forthright manner. As a result, your paper is now deemed suitable for publication.

Best wishes,

Reviewer 1 ·

Basic reporting

No comment

Experimental design

No comment

Validity of the findings

No comment

Additional comments

No comment

Cite this review as